# Blood-Based EWAS of Asthma Polygenic Burden in The Netherlands Twin Register

**DOI:** 10.3390/biom15020251

**Published:** 2025-02-08

**Authors:** Austin J. Van Asselt, René Pool, Jouke-Jan Hottenga, Jeffrey J. Beck, Casey T. Finnicum, Brandon N. Johnson, Noah Kallsen, Sarah Viet, Patricia Huizenga, Eco de Geus, Dorret I. Boomsma, Erik A. Ehli, Jenny van Dongen

**Affiliations:** 1Avera McKennan Hospital & University Health Center, Sioux Falls, SD 57105, USA; austin.vanasselt@avera.org (A.J.V.A.); jeffrey.beck@avera.org (J.J.B.);; 2Department of Biological Psychology, Vrije Universiteit, 1081 BT Amsterdam, The Netherlands; r.pool@vu.nl (R.P.);; 3Amsterdam Public Health Research Institute, 1081 HV Amsterdam, The Netherlands; 4Amsterdam Reproduction and Development (AR&D) Research Institute, 1081 HV Amsterdam, The Netherlands; 5Complex Trait Genetics, Center for Neurogenomics and Cognitive Research, Vrije Universiteit, 1081 HV Amsterdam, The Netherlands

**Keywords:** DNA methylation, asthma, polygenic risk, EWAS

## Abstract

Asthma, a chronic respiratory condition characterized by airway inflammation, affects millions of individuals worldwide. Challenges remain in asthma prediction and diagnosis from its complex etiology involving genetic and environmental factors. Here, we investigated the relationship between genome-wide DNA methylation and genetic risk for asthma quantified via polygenic scores in two cohorts from the Netherlands Twin Register; one enriched with asthmatic families measured on the Illumina EPIC array (n = 526) and a general population cohort measured on the Illumina HM450K array (n = 2680). We performed epigenome-wide association studies of asthma polygenic scores in each cohort with results combined through meta-analysis (total samples = 3206). The EWAS meta-analysis identified 63 significantly associated CpGs, (following Bonferroni correction, α = 0.05/358,316). An investigation of previous mQTL associations identified 48 mQTL associations between 24 unique CpGs and 48 SNPs, of which two SNPs have previous associations with asthma. Enrichment analysis using the 63 significant CpGs highlighted previous associations with ancestry, smoking, and air pollution. A dizygotic twin within-pair analysis of the 63 CpGs revealed similar directional effects between the two cohorts in 33 of the 63 CpGs. These findings further characterize the intricate relationship between DNA methylation and genetics relative to asthma.

## 1. Introduction

Asthma is a chronic and heterogeneous respiratory disease affecting approximately 262 million individuals globally that is characterized by variable airway obstruction, bronchial hyperresponsiveness, and inflammation [1,2]. Despite its substantial societal burden, predicting asthma risk and properly treating the illness remain significant challenges due to its complex etiology driven by the mixed genetic and environmental influences [3,4]. As a solution to accurately quantify asthma risk and allocate causes, disease risk scores have become of particular interest [5,6].

Specifically, polygenic risk scores (PRSs) have emerged as a potential tool to quantify genetic liability by aggregating the effects of multiple single-nucleotide polymorphisms (SNPs) identified through genome-wide association studies (GWASs) [7]. To date, GWASs have identified numerous loci associated with asthma susceptibility, including key genes that are implicated in immune regulation and airway inflammation. The individual effect sizes of these genetic variants, however, are modest, explaining a small proportion of the disease heritability [7]. Polygenic risk scores address this limitation by summing the weighted contributions of hundreds to thousands of risk alleles across the genome, improving the ability to estimate an individual’s genetic predisposition to asthma [8]. Recent advances in genomics and laboratory technology have enabled the development of these polygenic scores (PGS), allowing for the genotyping of large amounts of individuals in a cost-effective and high-throughput manner [9]. These scores have shown promise in enhancing our ability to predict asthma susceptibility, offering a potential tool for early intervention and personalized treatment strategies.

Recent studies have demonstrated the utility of PRSs in stratifying individuals by asthma risk. Several large-scale cohorts such as the UK Biobank and the Trans-National Asthma Genetic Consortium (TAGC) have enabled the development of PRS models of asthma liability in European populations [5,7,10,11]. These models incorporate variants from both common and low-frequency alleles, as well as loci associated with overlapping traits such as atopy and allergic rhinitis [10]. While the predictive power remains moderate, these scores have shown promise in distinguishing high-risk individuals from the general population with the top decile of asthma PRSs being three to eight times more likely to develop asthma compared to those in the bottom decile [10]. In addition, PRSs have demonstrated efficacy in predicting specific asthma phenotypes, such as allergic asthma or severe eosinophilic asthma, which are associated with distinct clinical trajectories and therapeutic responses [10].

Despite these advances, the clinical implementation of asthma PRSs has been limited. Currently, PRSs for asthma explain only a fraction of disease variation, suggesting that rare variants, gene–gene interactions, and epigenetic factors must be incorporated into future models to enhance predictive power [12,13]. Functional validation of PRS-associated loci is another critical challenge, as many GWAS hits map to non-coding regions, complicating the identification of causal variants and their biological mechanisms [14]. To mediate these shortcomings, additional scoring methods have been employed to assess asthma risk, including risk scores based on phenotypic characteristics, environmental exposures, and epigenetics [15].

DNA methylation and other epigenetic mechanisms play a significant role in asthma [16]. Specific methylation patterns are associated with asthma severity, airway inflammation, and immune responses [17,18]. Additionally, it has been shown that these DNA methylation profiles associated with asthma also associate with specific asthma clinical markers such as eosinophil counts, immunoglobulin levels, and reaction to specific allergens [19]. It is widely recognized that epigenetic mechanisms may mediate both the effects of environmental triggers (such as allergens) and genetic predisposition to asthma [15]. The environmental influences that have been shown to associate with asthma are expansive and numerous [20] and have an effect throughout life, from very early childhood through adulthood [20,21]. Exposure to various air contaminants strongly associate with the presence of asthma [20]. Exploration of the microbiome as a mediator of these exposures and allergic diseases [22] found a lack of microbial diversity, whether due to specific medications and treatments or a lack of outside exposures associated with atopic sensitization [23]. Respiratory infections early in life can also contribute to the presence of asthma throughout childhood [22,24,25]. These exposures can alter the methylome, resulting differentially methylated CpGs related to the severity of the infection [25]. Finally, certain treatment mechanisms can also alter DNA methylation [26]. For example, corticosteroid use can shift the methylome, further complicating the picture [26].

An investigation integrating epigenomic influences, specifically cell-specific chromatin structures, with asthma polygenic risk highlighted five genomic variant clusters in asthma-relevant cell types [27]. These findings demonstrated that SNVs modulating epithelial cell biology and lymphocyte cell processes were the strongest mediators of childhood-onset asthma [27]. Other recent studies of other traits, such as alcohol use disorder, schizophrenia, and autism, have integrated disease polygenic risk with genome-wide DNA methylation data to identify CpGs that might mediate the genetic risk of the disease [28,29,30].

Here, we investigated the association of asthma polygenic risk to specific CpGs through epigenome-wide association studies of the asthma polygenic scores in two independent datasets from the Netherlands Twin Register. The first is a population enriched with asthmatic individuals measured on Illumina EPIC arrays (hereafter referred to as the “NTR asthma EPIC dataset” n = 526). The second is a general population cohort from the Netherlands Twin Register (NTR) measured on Illumina 450k arrays (hereafter referred to as “NTR biobank 450k dataset”, n = 2680). For each cohort, a whole blood sample was collected from which DNA methylation profiles were generated. Polygenic scores for all individuals were computed using GWAS summary statistics from Demenais et al. (PMID: 29273806), based on a leave-on-out meta-analysis without NTR [11]. A meta-analysis of these results was then performed (n = 3206), which was followed by a subsequent dizygotic twin comparison investigating the significantly identified CpGs.

## 2. Materials and Methods

### 2.1. Asthma EPIC Array Dataset Participants

The study participants and the enrollment process have been previously described [31,32,33]. Participants came from twin families (young adult twins and their parents) from the Netherlands [33]. To be eligible for the study, at least one member of the family had to have been diagnosed with asthma, indicated by a self-reported survey in the Netherlands Twin Register (NTR) collected in the early 1990s (Appendix A). During a visit to the Vrije Universiteit Medical Center (VUMC), blood samples and biological and clinical measures of asthma were collected from 425 people [34]. A subset of 233 people also participated in the NTR Biobank project between 2004 and 2008 in which blood and buccal samples were collected [35]. DNA was extracted at the time of sample collection or immediately prior to the assessment of methylation [36]. All samples were simultaneously measured on the Illumina Infinium EPIC v1 methylation array. In total, 375 of the 425 individuals had a sample available for methylation analysis, of which 341 produced high-quality Illumina EPIC Array data from the first time point, and 232 individuals produced high-quality blood Illumina EPIC Array data from the second time point.

### 2.2. NTR Biobank 450k Array Dataset Participants

Illumina 450k array data were available on blood samples from twins and a small group of family members of twins who participated in the Netherlands Twin Register biobank project [35,37]. The enrollment process for these individuals has been previously described [35,37]. During this enrollment, a whole blood sample and buccal sample were collected. Whole blood samples were then assessed for methylation using the Illumina HumanMethylation 450K array. For the current EWAS, we excluded individuals who were also part of the other (NTR asthma EPIC dataset) dataset to perform two independent EWASs with no overlap in participants.

### 2.3. Ethics

The study was approved by the Central Ethics Committee on Research Involving Human Subjects of the VU University Medical Centre, Amsterdam, an Institutional Review Board certified by the U.S. Office of Human Research Protections (IRB number IRB00002991 under Federal-wide Assurance- FWA00017598; IRB/institute codes, NTR 03-180).

### 2.4. Sample Collection and DNA Extraction

The procedures of whole blood sample collection have been previously described [31,35,37]. The genomic materials analyzed in this study have been extracted and assessed for DNA quantity, quality, purity, and individual identity. The DNA analyzed in this study was extracted from whole blood samples using the Zymo Quick DNA mini-prep Kit (Zymo Research, Irvine, CA, USA) [36]. Genomic material was quantified via the Invitrogen Qubit Broad-Range Fluorescent Assay (Carlsbad, CA, USA), and sample purity was assessed using standard absorbance metrics via a SpectraMax microplate reader (Molecular Devices, San Jose, CA, USA).

### 2.5. Sample Genotyping

The genotyping process has been previously described [9,32]. Briefly, samples were genotyped using the Affymetrix SNP6 and Illumina Infinium Global Screening Array (GSA) [38]. Genotype data were then used to verify sample identities through identity by descent (IBD) analysis and assessed based on sample microarray performance.

### 2.6. Summary Statistics for Polygenic Scores

The polygenic scores (PGSs) in the Netherlands Twin Register (NTR) sample were based on the discovery GWA/GWAMA of Demenais et al. for asthma [11,39] after removing the NTR. We retained variants for which the effect allele frequency (EAF) was 0.01 ≤ EAF ≤ 0.99. Variant EAF and effect sizes were aligned with the NTR reference for the 1000 genomes variants. Discovery variants that were not part of this reference were discarded.

### 2.7. Generation of Weighted Effect Sizes

The software packages LDpred (version 0.9.1) and SBayesR (GCTB version 2.03) were utilized to generate the weighted effect sizes for calculation of the polygenic scores, as described below. The processed summary statistics were taken as input for the LDpred 0.9 software [39]. For estimating the target LD structure, we (1) used a selection of unrelated individuals in the NTR sample and (2) selected a set of well-imputed variants in the NTR sample. The parameter ld_radius was set by dividing the number of variants in common (from the output of the coordination step) by 12,000. Note that for the coordination step we provided the median sample size as input value for N. For the LDpred step we applied the following thresholds for fraction of variants with non-zero effect (in addition to the default infinitesimal model): --PS = 0.5, 0.3, 0.2, 0.1, 0.05, 0.01.

The SBayesR method is implemented in the gctb software [40,41]. For computing the weighted effect sizes, we took the above processed summary statistics as input and applied the following arguments:

--sbayes R; --pi 0.95, 0.02, 0.02, 0.01; --gamma 0.0, 0.01, 0.1, 1; --chain-length 40,000; --burn-in 4000; --out-freq 10. Note that for practical and computational reasons, we applied the LD matrix as provided by the authors of Lloyd-Jones et al., which is based on 2.8 million common variants in a random selection of 50,000 individuals of the UK biobank sample [42].

### 2.8. Scoring Weighted Effect Sizes in the NTR Sample

We used the PLINK software package (version 2.0.0) for generating the PGSs by applying the --score option to the input weighted effect sizes and the genotype dataset [43,44]. For scoring genotype datasets, we use the entire NTR sample or a subset of (non)transmitted paternal or maternal alleles. The latter subsets were generated by taking data from trios in the NTR genotype dataset.

Of each set of weighted effect sizes, we calculated the NTR PGSs over all genotype datasets. The selection of the optimal threshold value of the fraction of variants with non-zero effect was performed as follows: for each of the two cohorts, individuals were assessed for population outlier status. Any individual that was deemed a Netherlands population outlier was removed prior to further steps. This resulted in the removal of 15 and 132 individuals from the asthma-enriched cohort and the general population cohort, respectively. Next, individuals that were genotyped on the Affymetrix axiom platform were also removed to limit platform variability, which resulted in 12 (asthma cohort) and 85 (general population cohort) being removed from the two cohorts. To correct the polygenic scores for population stratification and genotype platform used, a linear model (using the lm() r function) was used with several covariates. This linear model corrected for the first 10 population-based principal components (PCs) (generated via the 1K genome project). Additionally, two genotype platforms, the Affymetrix SNP6 and Illumina GSA, were also included in this. The residuals were saved and used in subsequent analyses. These seven scores’ performance to predict asthma diagnosis was assessed in the asthma-enriched cohort (Appendix A). Based on these results, the best performing polygenic score was chosen for the rest of the analyses, which was the score based on the infinite fraction.

### 2.9. DNA Methylation Profiling

DNA bisulfite conversion was performed utilizing the Zymo EZ-96 DNA Methylation Kit [45]. DNA methylation was assessed using the Illumina Infinium EPIC v1 DNA Methylation Array on all samples at the Avera Genetics Laboratory [46]. The samples were fully randomized across arrays.

### 2.10. DNA Methylation Data Quality Control

Quality control has been described in detail previously [32,37]. Briefly, DNA methylation data quality was assessed by two bioinformatic tools. First, the preliminary DNA methylation quality assessment was completed via the Illumina GenomeStudio 1.0 software. Following this, methylation data quality control and normalization were completed for each dataset using the Biobank-based Integrative Omics Study (BIOS) Consortium pipeline in R. Sample quality was assessed using the R package MethylAid (v1.38.0) to omit any performance outliers (default thresholds) [47]. Additionally, sample identity was confirmed using the R package omicsPrint (to identify sex mismatches and mismatches between DNA methylation and genotype data). Array probe filtering and functional normalization were performed using the R package DNAmArray (v2.0.0) [48]. The following probe filters were applied: Probes were set to missing (NA) in a sample if they had an intensity value of exactly zero, detection *p*-value > 0.01, or bead count < 3. DNAmArray will also remove any probes that show a success rate below 0.95 across all samples. Finally, polymorphic probes, cross-reactive probes, and probes on the X and Y chromosomes were removed, which left a total of 742,442 CpGs for the asthma EPIC array dataset and 411,169 CpGs for the NTR biobank 450K array dataset. Finally, we calculated estimated cellular proportions for the Illumina EPIC array samples using the IDOL whole blood reference library [49].

### 2.11. EWAS

An EWAS of asthma polygenic score (PGS) was carried out in each of the two datasets separately, followed by meta-analysis of the two EWAS results. For each of the two EWASs performed, the r package gee (v4.13-25) was used due to the related nature of the samples, which corrects for the correlation structure in families. The following settings were utilized: Gaussian link function, 100 iterations, and the “exchangeable” option to correct for correlations within families and within persons (for 175 individuals, longitudinal blood samples were included). DNA methylation beta-values were regressed on residualized PGSs. Furthermore, age, sex, methylation array row, bisulfite sample plate (dummy-coded), smoking status (codes as 0 = Never smoker, 1 = former smoker, 2 = current smoker), and estimated cellular proportions (dataset 1, NTR asthma EPIC dataset), or measured cellular proportions (dataset 2, NTR biobank 450k dataset) were used as covariates. The R package Bacon (v1.32.0) was used to compute the Bayesian inflation factor and to obtain bias- and inflation-corrected test statistics prior to meta-analysis [50].

### 2.12. Meta-Analysis

Fixed-effects meta-analyses were performed on the overlapping CpGs of the Illumina 450k and EPIC array in METAL [51]. We used the *p*-value-based (sample size-weighted) method because of the differences between datasets. Statistical significance was assessed considering Bonferroni correction for the number of sites tested (number of overlapping CpGs = 358,316, α = 0.05/358,316). Additionally, we assessed if any CpG reached epigenome-wide significance in the individual EWAS at the array-specific Bonferroni threshold (EPIC dataset: α = 0.05/742,442, 450k dataset: α = 0.05/411,169) The I^2^ statistic from METAL was used to describe heterogeneity.

### 2.13. Follow-Up Analyses

Follow-up analyses were performed on significant CpGs from the meta-analysis and on top CpGs from each of the individual EWASs. An enrichment analysis was performed, querying the EWAS Atlas with the EWAS Toolkit (found online at https://ngdc.cncb.ac.cn/ewas/toolkit accessed on 17 December 2024), which compares against previously known associations from previous studies investigating DNA methylation. For the asthma-enriched cohort, the 100 most significant CpGs were utilized for this query, while the 95 significantly associated CpGs identified for the general population cohort were utilized for the second query. An additional assessment of the BBMRI mQTL database (based on the results from Bonder et al. [52]) was also performed utilizing these same CpGs.

### 2.14. Within Dizygotic Twin Pair Analysis

Finally, a comparison of dizygotic (DZ) twin pairs was performed in each of the two cohorts comparing adjusted beta values of the significantly identified CpGs from the meta-analysis to differences in asthma polygenic risk. For this comparison, residual beta values were calculated correcting for technical batch effects (ample plate and array row), sex, smoking status, and sample cellular proportions. These residual values were then analyzed to look for differences in DNA methylation within twin pairs in relation to differences in asthma PGS at each of the significantly identified CpGs. In total, 41 complete twin pairs were available in the NTR asthma EPIC dataset, and 394 complete twin pairs in the NTR biobank 450k dataset. Finally, a meta-analysis was performed using the results from the two cohorts.

## 3. Results

### 3.1. EWAS Descriptives and Meta-Analysis

#### 3.1.1. EWAS Descriptives

Demographic information of the cohorts is provided in Table 1. Genome-wide EWAS test statistics showed no inflation in the NTR asthma EPIC dataset and weak inflation in the NTR biobank 450k dataset, which was corrected for prior to the meta-analysis (Appendix A). One significant DMP (differentially methylated position) (α = 0.05/742,442) was detected in the NTR asthma EPIC dataset (cg26272069, *GABBR1* gene), and 95 significant DMPs were detected in the NTR biobank 450k dataset (α = 0.05/411,169) (Appendix A).

#### 3.1.2. Meta-Analysis

Meta-analysis of the two datasets (n = 3206) detected 63 significant DMPs over a total of 358,316 CpGs (α = 0.05/358,316) (Figure 1). Summary statistics for genome-wide methylation sites are provided in Appendix A. The top 20 significant sites from the meta-analysis are shown in Table 2. These 63 significant CpGs overlap with 58 CpGs from the NTR biobank 450k dataset, while the lone significant CpG from the NTR asthma EPIC dataset was not significantly associated in the meta-analysis. A total of 19 of these 63 CpGs showed opposite directions of effect. Two of the 63 DMPs (cg00570469; I^2^ = 97.6, and cg01200150 I^2^ = 97.2) displayed significant (α = 0.05/358,316) heterogeneity.

#### 3.1.3. EWAS Comparison

The singular CpG identified in the EWAS of the NTR asthma EPIC dataset, while present on the 450K array, was not significant in the EWAS of the NTR biobank 450k dataset; thus, no overlap in genome-wide significant CpGs was identified. This CpG, however, had the same direction of effect in each EWAS and did not show significant heterogeneity in the meta-analysis. A comparison of the estimates and *p*-values of the CpGs identified in the NTR biobank 450k dataset was performed comparing the two cohorts. This comparison included 80 CpGs (15 of the 95 CpGs were exclusive to the 450K array). This comparison yielded a correlation of 0.13 for the estimates between the two cohorts.

### 3.2. EWAS Trait Enrichment Analysis

An enrichment analysis was performed using the EWAS Atlas (https://ngdc.cncb.ac.cn/ewas/atlas/index accessed on 17 December 2024) to identify any enrichment for previous known associations of the CpGs identified in the meta-analysis. This identified four CpGs with previous associations to ancestry and air pollution that are present in the meta-analysis. The results from this analysis are displayed in Figure 2.

In order to examine any differences between the EWAS in the NTR asthma EPIC dataset and the EWAS in the NTR biobank 450k dataset, we additionally performed enrichment analyses for each of these EWAS separately. This showed that the top 100 CpGs from the NTR asthma EPIC dataset were shown to mostly be previously associated with fractional exhaled nitric oxide (FENO), allergic sensitization, and allergic asthma with 21, 7, and 5 previous associations, respectively (Appendix A). By contrast, the 95 significantly associated CpGs from the NTR biobank 450k dataset EWAS were enriched for associations with ancestry, smoking, and air pollution based on 6, 6, and 4 overlapping CpGs, respectively (Appendix A).

### 3.3. Comparison to Previously Known mQTL Associations

An investigation of the BBMRI mQTL database based on the results by Bonder et al. (PMID: 27918535) was performed to assess for any overlap with previously identified mQTL associations [52]. The 63 significant CpGs from the meta-analysis were found to have 48 mQTL associations in total (41 cis- and 7 trans-). The 7 trans-mQTL associations were from a single CpG, cg14663208. Two of these seven mQTL associations were with SNPs (rs2523722 and rs2021722) that were previously shown to be associated with asthma by Demenais et al. (PMID: 29273806) [11]. A detailed list of these mQTL associations can be found in Appendix A.

Next, independent investigations of each EWAS were performed, searching for previous mQTL associations. Six previous mQTL associations were identified for the single CpG identified in the EWAS of the NTR asthma EPIC dataset (Appendix A). None of the six SNPs that were associated with an mQTL were SNPs identified by Demenais et al. [11] to be significantly associated with asthma. Investigating the 95 significantly associated CpGs from the NTR biobank 450k dataset EWAS highlighted 55 and 7 cis- and trans-mQTL associations, respectively (Appendix A).

### 3.4. Overlap with Previous EWASs of Asthma Clinical Markers

In a previous study, we performed a principal component-based EWAS of asthma clinical markers on the same asthma cohort measured on the EPIC array [19]. This study found 221 unique, significant CpG associations with various asthma clinical markers [19]. Here, we investigate the overlap between those CpGs and the CpGs associated with asthma polygenic risk in our current analyses. The 63 significant CpGs from the meta-analysis did not contain any overlap with the 221 CpGs identified in our previous study.

We also compared those results to the results from each individual EWAS. Like the EWAS Atlas enrichment analysis, the top 100 most-significant CpGs from the NTR asthma EPIC dataset EWAS were compared. We observed five overlapping CpGs out of these 100 with the 221 CpGs identified in our previous study (Table 3). Furthermore, a correlation analysis of the EWAS estimates (the estimates generated in the EPIC EWAS from this study and EWAS estimates from our previous study) for the top 100 CpGs yielded a very strong correlation (r = 0.995). The five overlapping CpGs were found to associate mostly with whole blood eosinophil counts and immunoglobulin levels in our previous study. Finally, a similar comparison of the 95 significantly associated CpGs identified in the NTR biobank 450k dataset was performed, but none of these CpGs were found to overlap with the CpGs identified in the EWAS of asthma clinical markers.

### 3.5. Dizygotic Twin Pair Comparison

Using the 63 significantly associated CpGs identified in the meta-analysis, a dizygotic (DZ) twin comparison was performed in each population cohort looking at differences in asthma polygenic risk and DNA methylation (41 twin pairs in the NTR asthma EPIC dataset and 394 twin pairs in the NTR biobank 450k dataset). From these two analyses, a meta-analysis was conducted, which showed that no CpGs reached statistical significance. In total, 30 of the 63 CpG estimates showed an opposite direction of effect in the two cohorts, while, conversely, 33 of the CpGs showed similar directional effects between the two cohorts. A summary of the meta-analysis can be found in Appendix A.

## 4. Discussion

The complexity of asthma stems from its heterogeneity in presentation as well as its susceptibility to be influenced by a myriad of internal stimuli (including genetic predisposition) and external stimuli. We investigated the epigenetic correlates of asthma polygenic risk through meta-analysis of two independent datasets from the Netherlands Twin Register: an asthma-enriched cohort measured with the Illumina Infinium EPIC array and a general population cohort measured with the Illumina Infinium 450K array. The meta-analysis of the two EWAS identified 63 significantly associated CpGs. These CpGs were significantly enriched for CpGs that were previously found to be associated with ancestry, air pollution exposure, and smoking [53]. In total, 24 of the CpGs were associated with 48 mQTLs (of which two SNPs have been previously identified through a GWAS of asthma). Of note, both air pollution and smoking have been previously shown to dramatically increase both risk and severity of asthma [54,55]. The overlap of CpGs associated with asthma PGS and CpGs associated with air pollution could have multiple potential explanations, including (1) an increased genetic propensity for asthma risk acts through exposure to these environments via DNA methylation, and (2) asthma genetic risk and smoking or air pollution act independently on the same epigenetic pathways. Overlap with previous CpGs relating to ancestry could hint at this association of differential methylation to genetic risk of asthma being ancestrally specific, which could potentially be related to genetic differences or environmental differences between individuals of different ancestry. Additional studies investigating this interaction in cohorts of different ancestries could further the understanding of this potential link.

As presented in Table 2, many of the genes that these significantly associated CpGs reside in have not been previously linked to asthma. This, combined with the identified overlap with environmental exposures, hints that this epigenetic signature may be environmentally linked. Environmental exposures to air pollutants, smoking, specific asthma treatments, and respiratory viral infections all appear to play a critical role in the development of asthma [24,25]. It is possible that an individual’s propensity to be exposed to these environmental effects could be genetically linked, which is captured within the asthma polygenic score. As an example, the overlap we identified with air pollution exposure could point to genetically driven behaviors that influence the methylome and the presence of asthma. Future studies incorporating detailed records of these environmental stimuli could lead to a better understanding of this potential link.

Our meta-analysis uncovered some differences in the EWAS results between the two cohorts, which may reflect differences in the epigenetic landscape of individuals with varying levels of expression of asthma. Of the 63 CpGs, 58 CpGs were also genome-wide significant in the EWAS of the NTR biobank 450k dataset, while the single genome-wide significant CpG from the NTR asthma EPIC dataset did not reach significance in the meta-analysis. Although the meta-analysis detected some evidence for heterogeneity, only two CpGs showed significant heterogeneity, and 44 CpGs were shown to have the same directional association in each cohort. The EWAS conducted in the NTR asthma EPIC dataset identified a single significant CpG site, cg26272069, after Bonferroni correction. Cg26272069 has been identified to be previously associated allergic sensitization [56]. Though this association has been recorded, the underlying biological mechanism remains unknown. This CpG was also shown to have six previously identified cis-mQTL associations. While none of these SNPs overlap with the SNPs identified in the asthma genome-wide association study (GWAS) by Demenais et al. (from which the polygenic scores used here were derived from), they highlight the potential relevance of cg26272069 in epigenetically mediated genetic risk of asthma [11]. The enrichment analysis of the top 100 CpGs from the asthma-enriched cohort revealed additional prior associations with asthma-specific traits such as fractional exhaled nitric oxide (FENO) and allergic sensitization. This alignment with known physiological markers of asthma hint at the possibility that asthma polygenic risk could be influencing these changes in DNA methylation that associate with asthma.

The EWAS performed in the general population cohort yielded 95 significantly associated CpGs. When investigating these CpGs for any previous mQTL associations, 55 cis-mQTL, and 7 trans-mQTL associations were found. Interestingly, two of these trans-mQTL associations included SNPs previously linked to asthma in the GWAS by Demenais et al. [11]. This observation suggests that the genetic burden for asthma may be epigenetically regulated through the direct influence of specific risk alleles. However, the 95 significant CpGs identified here had no overlap with previous EWASs of asthma. A query of the EWAS Atlas via an enrichment analysis showed that the most abundant previous associations were with specific environmental exposures such as smoking and air pollution.

We further compared our findings directly to our previous study investigating the association of DNA methylation to asthma clinical markers based on the same asthma cohort as included in the current meta-analysis [19]. This comparison, similar to the results found in the EWAS Atlas enrichment analysis, showed a distinct difference in overlap between the results from the two cohorts. Notably, six of the top asthma PGS-associated CpGs from the NTR asthma EPIC dataset overlapped with CpGs previously identified in our previous EWAS of asthma clinical markers. These CpGs were strongly correlated with markers such as eosinophil counts and immunoglobulin levels. On the other hand, none of the significant CpGs from the PGS-EWAS in the NTR biobank 450k dataset were associated with asthma clinical markers. The findings suggest that DNA methylation associated with asthma polygenic risk in a population cohort may capture a different signal compared to the signal identified in a cohort enriched with people with an asthma diagnosis. This could be related to differences in the frequency of asthma and underlying symptoms, differences in DNA methylation, and differences in allele frequencies.

Importantly, the causal chain of events that explains the associations identified in this work remains to be determined. It could be that asthma polygenic risk causes changes in DNA methylation, and, subsequently, these DNA methylation alterations lead to specific physiological changes and asthmatic symptoms. Alternatively, it is possible that asthma itself or underlying physiological changes in asthma cause changes in DNA methylation. A third explanation can be that the PGS influences DNA methylation and asthma independently (horizontal genetic pleiotropy). Future studies are warranted to distinguish between these alternative explanations. Associations between any trait and PGS can also be inflated by population stratification and assortative mating. Within-family designs are gaining popularity in PGS research as they offer better control for confounding. Here, we leveraged data from dizygotic twin pairs to test if within-pair differences in DNA methylation are predicted by within-pair differences in asthma polygenic risk. Dizygotic twin pairs share a significant portion of their external environment throughout their life, including fetal development, as well as 50% of their genetic background, and have the same age. Furthermore, because twins have the same parents, within-pair associations cannot be biased by population stratification or assortative mating. For this reason, specifically looking within DZ twin pairs for differences in methylation in relation to a particular trait can omit confounders in the general population while still capturing influence from the genome through polygenic risk (which would not be possible with monozygotic twins). Here, we found that none of the significant CpGs identified in the meta-analysis were significantly associated with changes in asthma polygenic risk in the within-pair analysis in DZ twins. In addition to not identifying significant CpGs, the estimates of the CpGs were found to vary in direction between the two cohorts (30 of 63 had different directions between the two cohorts). When comparing these results from the DZ twin analysis to those of the meta-analysis of the two EWASs, 22 out of the 63 significant CpGs were identical in direction. The lack of significant findings and limited directional similarity could be for several reasons. The relatively small sample size could be playing a role in identifying significant loci, and including a greater number of twin pairs could produce significant findings. Furthermore, it could be the case that some of the significant associations identified in the meta-analysis of the two EWASs could be capturing influences controlled in the within-pair analysis (such as fetal development and similar environmental exposures). Additional future studies utilizing large quantities of twin pairs could provide added clarity to these findings and the nuances influencing asthma and its presentation.

This study does contain some limitations that are important to consider. First, both cohorts were assessed on different DNA methylation array platforms at different times. Ideally, having all sample data generated on the same platform would produce the most comparable data. The results in this manuscript are all derived from a Dutch population of European ancestry, which can limit the scope of these findings. Additional research is needed to broaden these results and confirm their applicability across other ancestries. While our dataset contains some variables relating to environmental exposures (such as smoking), the utilization of a dataset more rich in characterizing a broad range of exposures could provide better insight into which environmental factors are driving these associations. Future studies including data regarding air pollution exposure, previous respiratory infections, and other variables could determine which specific exposures relate to asthma. The use of whole blood samples, while providing several benefits, has some drawbacks, notably, in regard to the cell-type heterogeneity they possess (though this can be managed through bioinformatic processes to an extent) [57]. The utilization of single-cell methods, or methods utilizing cell sorting techniques to investigate distinct populations of white blood cells, could provide some additional clarity on the nuances between DNA methylation and asthma polygenic risk. Furthermore, the investigation of other epigenetic modifications (though not possible here due to sample limitations) could also provide a more comprehensive understanding of the epigenome as it relates to asthma polygenic risk. Future studies that investigate multiple epigenetic mechanisms in relation to asthma and asthma polygenic risk could uncover novel interactions not yet known. Finally, an increase in sample size, especially an increase in the number of complete dizygotic twin pairs, could improve the statistical power of this study, and it offers the possibility to exclude confounding and an increased ability to glean more information on the interplay between DNA methylation and asthma polygenic risk.

## 5. Conclusions

Asthma is a complex disease that is heavily influenced by both genetics and the environment. This study provides novel insights into the interplay between genetic predisposition and epigenetic regulation through DNA methylation. Through meta-analysis of epigenome-wide association studies across two independent Dutch cohorts, we identified 63 CpGs significantly associated with asthma polygenic risk. Notably, 48 of these CpGs were linked to mQTL associations, involving 24 unique CpGs and 48 SNPs. These 63 CpGs have previously documented associations with ancestry, smoking, and air pollution exposure. Together, these results underscore the importance of considering both genetic and epigenetic factors, and their interplay, to better understand asthma etiology.

## Figures and Tables

**Figure 1 biomolecules-15-00251-f001:**
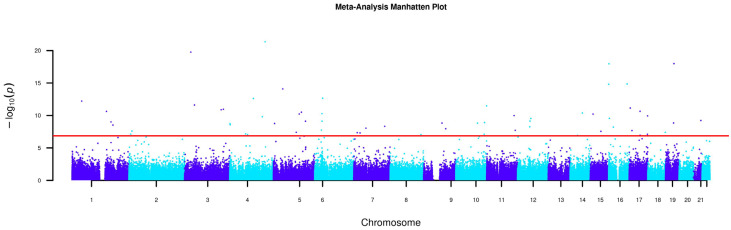
Manhattan plot of the meta-analysis results from the two EWASs.

**Figure 2 biomolecules-15-00251-f002:**
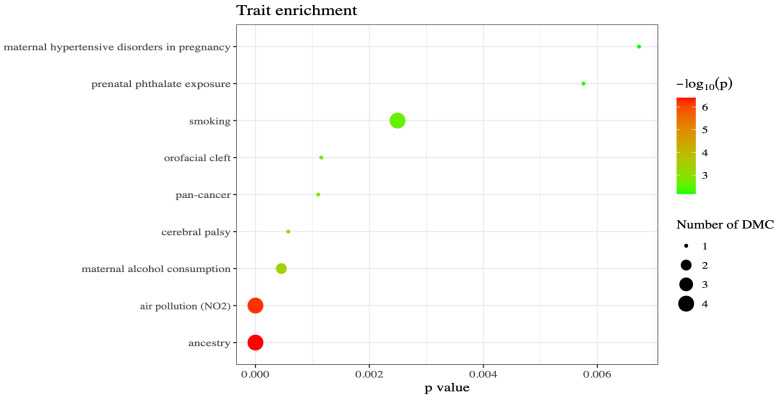
Results from the enrichment analysis of the 63 significant CpGs from the meta-analysis. The plot shows the number of previously identified Differentially Methylated CpGs (DMC) that are present within the CpGs identified in our meta-analysis. The EWSA Atlas job ID for the complete analysis is: 331c3fd45b66606f6e860a6779ce594c.

**Table 1 biomolecules-15-00251-t001:** Demographic information for both cohorts assessed in the two epigenome-wide association studies.

Cohort	Participants	CpGs	Average Age	Age Range	Males	Females
EPIC	526	742,442	40.1	13–77	241	285
450K	2680	411,169	36.5	18–80	894	1786

**Table 2 biomolecules-15-00251-t002:** Summary statistics for the 20 most significant CpGs from the meta-analysis. The Direction column depicts the direction of effect within each cohort, with the first (+/−) symbol reflecting the 450K cohort and the second symbol reflecting the EPIC cohort.

CpG	Z Score	*p* Value	Direction	HetPVal	Chr	Position	Gene
cg02398359	9.66	4.44 × 10^−22^	++	0.07305	4	154541136	KIAA0922
cg21032395	9.28	1.73 × 10^−20^	++	0.3174	3	27528777	
cg00570469	−8.83	1.04 × 10^−18^	−+	7.79 × 10^−11^	19	36602113	
cg19605773	8.82	1.11 × 10^−18^	++	0.04192	16	1435582	
cg06919916	7.99	1.38 × 10^−15^	−−	6.28 × 10^−5^	16	80838738	CDYL2
cg10010898	7.97	1.54 × 10^−15^	−−	0.000357	16	339598	AXIN1
cg04002187	−7.77	8.10 × 10^−15^	−+	1.57 × 10^−5^	5	40835754	RPL37
cg04690793	7.34	2.18 × 10^−13^	++	0.1864	6	34723390	SNRPC
cg18277508	−7.32	2.43 × 10^−13^	−+	2.11 × 10^−6^	4	103266345	SLC39A8
cg14663208	−7.20	6.03 × 10^−13^	−−	0.2682	1	42367407	HIVEP3
cg00410650	7.01	2.48 × 10^−12^	++	0.1318	3	43404346	
cg09893465	−6.97	3.29 × 10^−12^	−−	0.2296	10	134588040	INPP5A
cg18750756	−6.86	7.08 × 10^−12^	−−	0.006945	17	4337231	SPNS3
cg02062140	6.79	1.10 × 10^−11^	++	0.04255	3	169781664	GPR160
cg06698292	6.77	1.33 × 10^−11^	++	0.4791	3	160113492	IFT80
cg24238265	−6.69	2.27 × 10^−11^	−−	0.04164	17	47208515	B4GALNT2
cg01139966	−6.68	2.37 × 10^−11^	−−	0.02477	1	150669796	GOLPH3L
cg05389183	−6.64	3.11 × 10^−11^	−−	0.01427	5	122372942	PPIC
cg20029311	−6.60	4.22 × 10^−11^	−−	0.07963	14	74100086	
cg24536200	6.56	5.27 × 10^−11^	++	0.006486	6	31707961	MSH5

**Table 3 biomolecules-15-00251-t003:** Summary of the five overlapping CpGs from the NTR asthma EPIC dataset PGS EWAS and the previous EWASs of asthma clinical markers based on the same cohort. The estimate and *p*-value columns represent the results generated in this analysis associating with asthma polygenic risk, while the “Original Estimate” and “Original *p*-value” columns report the previously published EWAS results associating with asthma clinical marker derived principal components.

CpG	Estimate	*p*-Value	Original Estimate	Original *p*-Value	Chr	Position
cg01284182	−0.004978495	6.56 × 10^−6^	−0.00868249	1.30 × 10^−8^	15	39823154
cg02068052	−0.004662464	6.01 × 10^−7^	−0.007940227	5.15 × 10^−8^	19	15436097
cg16270995	−0.007554705	1.55 × 10^−6^	−0.012109812	3.46 × 10^−8^	14	77425779
cg18058448	−0.003540704	2.85 × 10^−6^	−0.006594822	5.38 × 10^−9^	20	33049021
cg20178107	−0.005601211	1.81 × 10^−6^	−0.009781225	1.38 × 10^−8^	7	41914145

## Data Availability

The HumanMethylation450 BeadChip data from the NTR are available as part of the Biobank-based Integrative Omics Studies (BIOS) Consortium in the European Genome-phenome Archive (EGA), under the accession code EGAD00010000887. They are also available upon request via the BBMRI-NL BIOS consortium (https://www.bbmri.nl/acquisition-use-analyze/bios accessed on 30 December 2024). All NTR data can be requested by bona fide researchers (https://ntr-data-request.psy.vu.nl/ accessed on 30 December 2024). Because of the consent given by study participants the data cannot be made publicly available. The pipeline for DNA methylation-array analysis developed by the Biobank-based Integrative Omics Study (BIOS) consortium is available here: https://molepi.github.io/DNAmArray_workflow/ (https://doi.org/10.5281/zenodo.3355292) (accessed on 30 December 2024).

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
