# Peer review of "Blood-Based EWAS of Asthma Polygenic Burden in The Netherlands Twin Register"

_biomolecules, 2025, doi:10.3390/biom15020251_

Round 1

Reviewer 1 Report

Comments and Suggestions for Authors

This study performed epigenome-wide association studies of asthma polygenic scores in two cohorts with results combined through meta-analysis. The analysis procedure is standard and straightforward. 

My major comment is about the motivation for meta analysis. It appears that individual level data are available, it that right? If so, have you considered pooling the two samples for a single epigenome-wide association study? I understand pooling can reduce the number of CpG sites due to the difference in methylation array.

Minor comments: 

1. Add "DMP" to the Abbreviations list

2. Sample sizes do not match: 526+2680 is 3206, not 3209. This inconsistency appears in multiple places, including the abstract. 

3. Remove "2.6.1. Summary statistics" since it is the only one subsection. Do the same for "2.7.1. LDpred v0.9".

4. It would be nice to state the purpose of Sections "2.7. Generation of weighted effect sizes" and "2.8. SBayesR". Right now they just describe what were done and no word on what are for. 

Author Response

This study performed epigenome-wide association studies of asthma polygenic scores in two cohorts with results combined through meta-analysis. The analysis procedure is standard and straightforward.

My major comment is about the motivation for meta analysis. It appears that individual level data are available, it that right? If so, have you considered pooling the two samples for a single epigenome-wide association study? I understand pooling can reduce the number of CpG sites due to the difference in methylation array.

• Thank you very much for the feedback and comments that you provide on our manuscript. The main motivation for conducting a meta-analysis (instead of a single EWAS) was in fact due to the multiple differences between the two data sets. As you mention, combining the two data sets would have resulted in a significant number of CpGs being left out of the analysis entirely. Along with the differences in measured CpGs, each cohort was measured on the methylation arrays independently at different times, which could have made correcting for technical batch effects more difficult.

Minor comments:
1. Add "DMP" to the Abbreviations list
• DMP has been added to the Abbreviations list, and it has also been clarified what the abbreviation means within the manuscript text when it first appears.
2. Sample sizes do not match: 526+2680 is 3206, not 3209. This inconsistency appears in multiple places, including the abstract.
• This has been corrected and now reflects the correct sample sizes across the entire manuscript (526, 2680, 3206).
3. Remove "2.6.1. Summary statistics" since it is the only one subsection. Do the same for "2.7.1. LDpred v0.9".
• The methods section was updated by removing these additional section headers for added clarity.
4. It would be nice to state the purpose of Sections "2.7. Generation of weighted effect sizes" and "2.8. SBayesR". Right now they just describe what were done and no word on what are for.
• These two methods were both utilized to generate the weighted effect sizes that were used in this analysis. The headers have been restructured to better state this. Additionally, a statement has been added to clarify this at the beginning of this sub-section of the methods section.

Reviewer 2 Report

Comments and Suggestions for Authors

This is a study investigating the association of blood DNAm with asthma PRS. Overall, this is an interesting study. However, the clarity of the cohort demographics and literature review needs to be largely improved. I have a few comments:

1.       Table 1 only shows age and sex for demographic characteristics of the participants. Please add additional information if avaiable, such as environmental factors (e.g., air pollution level, reent respiratory infection, corticosteroid use, diet), age range (not just average), race/ethnicity, etc…

2.       Table 2: most of the genes are not well-known as asthma genes, such as HLA, 17q1-21, FLG, etc… This highlights the relationship between DNAm and asthma PRS is potentially more environmental factor related. Thus, it is important discuss more on the environmental factors, see my following comments #3-5.

3.       The relationship between blood DNAm and asthma PRS can show how asthma genetic risk interplay with the environmental risk factors by DNAm. However, the environmental data is not available in this study, this needs to be added as a limitation.

4.       The authors discussed smoking and air pollution as the major environmental exposure. However, some additional environmental risk factors, such as respiratory infection and microbiome, have strong effect for asthma, especially childhood asthma. Given the NTR cohort has young adults, it is important to discuss some childhood related environmental risk factors and their interplay with asthma genetic risk through PRS or Mendelian randomization. Please make a discussion based on these papers PMID: 39326916, 37679381.

5.       Another important environmental factor for asthma is the IgE sensitization and related corticosteroid use, it is important to discuss how this can interplay with epigenetics (DNAm and non-coding RNAs) and its mechanisms (e.g., Fc epsilon receptor) for asthma. Here are some papers to consider PMID: 31187518.

Author Response

This is a study investigating the association of blood DNAm with asthma PRS. Overall, this is an interesting study. However, the clarity of the cohort demographics and literature review needs to be largely improved. I have a few comments:

1. Table 1 only shows age and sex for demographic characteristics of the participants. Please add additional information if avaiable, such as environmental factors (e.g., air pollution level, reent respiratory infection, corticosteroid use, diet), age range (not just average), race/ethnicity, etc…

• We have expanded Table 1 to include the age range of individuals in each cohort. We do not have available to us data regarding the listed environmental factors for all individuals within both cohorts. We do correct for asthma medication usage in each of the EWAS, which we have stated in the methods section. The lack of information regarding some of these environmental exposures has been added as a limitation within this study. Regarding race/ethnicity, we excluded any individuals that were ethnic outliers based on their genotype data, so the population consists of individuals of similar ethnic background from the Netherlands. This has been clarified in the methods section of the manuscript as well.

2. Table 2: most of the genes are not well-known as asthma genes, such as HLA, 17q1-21, FLG, etc… This highlights the relationship between DNAm and asthma PRS is potentially more environmental factor related. Thus, it is important discuss more on the environmental factors, see my following comments #3-5.

• As your comment points out, many of the genes we identified are not the traditional genes often associated with asthma. We have added a paragraph within the discussion section describing this and its potential implications. We have also added a comment on the need for additional environmental data within the limitations paragraph of our manuscript.

3. The relationship between blood DNAm and asthma PRS can show how asthma genetic risk interplay with the environmental risk factors by DNAm. However, the environmental data is not available in this study, this needs to be added as a limitation.

• As was mentioned in our above responses to comments 1 and 2, we have added this to our limitations section and stressed the importance of future studies to attempt to capture this type of data.

4. The authors discussed smoking and air pollution as the major environmental exposure. However, some additional environmental risk factors, such as respiratory infection and microbiome, have strong effect for asthma, especially childhood asthma. Given the NTR cohort has young adults, it is important to discuss some childhood related environmental risk factors and their interplay with asthma genetic risk through PRS or Mendelian randomization. Please make a discussion based on these papers PMID: 39326916, 37679381.

• This comment makes an excellent point bringing up some additional environment exposures that can contribute to asthma. To broaden our manuscript and make the introduction and discussion more well-rounded, we have added sentences to both describing these specific environmental risk factors and how they could be influencing our results. Additionally, we also reference the two manuscripts listed here to substantiate this added discussion topic.

5. Another important environmental factor for asthma is the IgE sensitization and related corticosteroid use, it is important to discuss how this can interplay with epigenetics (DNAm and non-coding RNAs) and its mechanisms (e.g., Fc epsilon receptor) for asthma. Here are some papers to consider PMID: 31187518.

• IgE sensitization and corticosteroid use are two hallmark phenomena typically associated with asthma. As is mentioned within this comment, the inflammatory pathways that these two mechanisms interact with could be epigenetically driven. We have added a paragraph within the introduction and discussion sections of this manuscript (along with the mentioned manuscript for reference) to address this point, and to ensure that we cover critical topics related to asthma and epigenetics.

Reviewer 3 Report

Comments and Suggestions for Authors

With real interest, I read the manuscript biomolecules-3430077 written by experienced researchers. The study is solid and the manuscript is well written.

Comments:

1.      Whole blood is a limiting factor in epigenetic/epigenomic studies, even though it can be a kind of bioinformatically managed (PMID: 28322581, 24495553). It is widely used as whole blood samples are widely stored in biobanks. To get biologically informative data, sorted cell populations if not single cells (where possible and reasonable) should be used. On the other hand, using whole blood samples for DNA methylation studies targeting biomarker identification can be of benefit as simplifies potential diagnostic method. However, it is also possible that DNA methylation analysis performed in sorted cells would be more informative also from the diagnostic point of view.

Anyway, please, discuss as a limitation.

2.      What about other epigenetic modifications, would there be applicable in such studies? For example, histone modifications, which are rarely studied in epigenomic context due to complex and demanding sample preservation and processing methodology.

Other comments:

1.      Supplemental Table 1, Supplemental Table 6, Supplemental Figure 1, and Supplemental Figure 2 should be removed from the main file. Supplemental Table 1 and Supplemental Table 6 should be moved to the supplement.

Author Response

With real interest, I read the manuscript biomolecules-3430077 written by experienced researchers. The study is solid and the manuscript is well written.

• Thank you very much for committing your time to read our manuscript and for the constructive feedback.

Comments:

1. Whole blood is a limiting factor in epigenetic/epigenomic studies, even though it can be a kind of bioinformatically managed (PMID: 28322581, 24495553). It is widely used as whole blood samples are widely stored in biobanks. To get biologically informative data, sorted cell populations if not single cells (where possible and reasonable) should be used. On the other hand, using whole blood samples for DNA methylation studies targeting biomarker identification can be of benefit as simplifies potential diagnostic method. However, it is also possible that DNA methylation analysis performed in sorted cells would be more informative also from the diagnostic point of view.
Anyway, please, discuss as a limitation.

• As you state in your comment, whole blood samples provide some great opportunities and advantages, but they do also have their drawbacks. Whole blood samples being widely available and easily obtainable through less invasive techniques make them prime candidates to utilize in the context of disease biomarkers. Additionally, whole blood samples are inherently linked to diseases of the immune system, such as asthma, which also makes these sample types a viable option. Though we can estimate the cellular composition of these samples (based on their methylation profiles), using cell-type specific populations could provide clarity about which cell types are contributing more to these associations we identified. The utilization of cells from a specific population of white blood cells, or even the use of single cell techniques, could provide insight into the nuances of how these methylation profiles differ from cell to cell. These thoughts and comments have been added to the limitations of our study to clearly state how future studies could benefit from using more specific populations of cells.

2. What about other epigenetic modifications, would there be applicable in such studies? For example, histone modifications, which are rarely studied in epigenomic context due to complex and demanding sample preservation and processing methodology.

• Epigenetic modifications outside of DNA methylation could absolutely be an area for future studies to explore in relation to asthma, more generally, and asthma polygenic risk, specifically. Due to the limitations of the samples that we had available to us (in terms of available DNA from the samples) we were only able to investigate DNA methylation via these array platforms, as they are able to function reliably with only a few hundred nanograms of DNA. To reflect this importance to also investigate these types of epigenetic modifications, we have added statements in the limitations sections of the manuscript.

Other comments:

1. Supplemental Table 1, Supplemental Table 6, Supplemental Figure 1, and Supplemental Figure 2 should be removed from the main file. Supplemental Table 1 and Supplemental Table 6 should be moved to the supplement.

• Supplemental Tables 1 and 6 along with Supplemental figures 1 and 2 will be exclusively placed within the supplemental section and not within the main text of the manuscript.

Round 2

Reviewer 2 Report

Comments and Suggestions for Authors

The authors have revised the paper well. I have no further comments.